# Applications of Metal-Organic Frameworks as Drug Delivery Systems

**DOI:** 10.3390/ijms23084458

**Published:** 2022-04-18

**Authors:** Bianca Maranescu, Aurelia Visa

**Affiliations:** 1Coriolan Dragulescu Institute of Chemistry, 24 Mihai Viteazul Blv., 300223 Timisoara, Romania; 2Department of Biology-Chemistry, Faculty of Chemistry, Biology, Geography, West University Timisoara, 16 Pestalozzi Street, 300115 Timisoara, Romania

**Keywords:** metal organic frameworks, drug delivery systems, surface modification, encapsulation, stimulus

## Abstract

In the last decade, metal organic frameworks (MOFs) have shown great prospective as new drug delivery systems (DDSs) due to their unique properties: these materials exhibit fascinating architectures, surfaces, composition, and a rich chemistry of these compounds. The DSSs allow the release of the active pharmaceutical ingredient to accomplish a desired therapeutic response. Over the past few decades, there has been exponential growth of many new classes of coordination polymers, and MOFs have gained popularity over other identified systems due to their higher biocompatibility and versatile loading capabilities. This review presents and assesses the most recent research, findings, and challenges associated with the use of MOFs as DDSs. Among the most commonly used MOFs for investigated-purpose MOFs, coordination polymers and metal complexes based on synthetic and natural polymers, are well known. Specific attention is given to the stimuli- and multistimuli-responsive MOFs-based DDSs. Of great interest in the COVID-19 pandemic is the use of MOFs for combination therapy and multimodal systems.

## 1. Introduction

It has been reported that DDS contains two main components: a carrier and a drug. The optimal properties of both components are precisely checked by careful selection of these constituents [1].

Conventional DDS are tablets, capsules, granules, and ointments; syrups for oral administration; and suppositories or solutions for intravenous administration. Owing to several disadvantages and various limitations, such as pure absorption for target sites, repeated dosage several times per day, high dose needed, fluctuations in plasma drug level, difficult to monitor, poor bioavailability issues, critical toxicities, side effects, and premature excretion from the body, conventional DSSs are unable to reach sustained release [2]. In this respect, new DDSs are required and being investigated.

Currently, numerous DDSs have been created to reduce side effects and increase therapeutic efficacy. Thus, inorganic materials such as carbon nanotubes [3,4], graphene [5], magnetic nanoparticles (NPs) of iron oxide [6,7], silicon-based materials and their oxides [8,9], and gold nanoparticles [10,11,12] were used.

Metal–organic frameworks (MOFs) are an interesting and versatile class of coordination polymers, assembled from metal ion/clusters and organic linkers. MOFs have diverse properties, such as high porosity and surface area, large pore volume for encapsulation of various substances/gazes, and special chemical and thermal stability [13,14,15,16]. They have been used in various applications, such as gas storage [17], photochemistry [18], catalysis [19,20,21], flame retardants [22,23,24], corrosion protection [25,26], separation processes [27], adsorption properties [28,29,30,31,32,33,34], diagnostic [35], antimicrobial properties [36,37,38], and delivery of a large variety of active drug molecules, cosmetics, and biological gases [39,40,41]. Owing to these characteristics, MOFs have become the most promising materials for use in biomedicine [42,43].

The characteristics, such as nontoxic effects of MOFs, directed and stimulus-based delivery systems, multiple drug loaded properties, and continuous release, have enriched the use of MOFs in drug delivery, biocompatibility, and biodegradability in the last decade. One of its most significant properties, which is continuously explored by scientists, is MOFs capability of interacting with biological systems based on various stimuli encountered inside our system, which includes precise and sustained drug release capacity and improved solubility of amorphous and poorly soluble drugs in variable conditions such as pH, temperature, light, magnetic, pressure, glucose level, multiple stimuli-responsive systems [44,45].

MOFs were originally synthesized by a method called solvothermal synthesis, which has been the most popular technique for MOFs synthesis. After two decades from the first MOFs reference, new improvements in synthesis approaches were accomplished. New methods, which involved the use of mechanochemical, electrochemical, sonochemical, and microwave-assisted technique, have been reported in literature with both advantages and disadvantages, as presented in Figure 1 [46,47,48,49].

Because of the difference in several orders of magnitude between the size of nano-MOFs and the size of cancer cells, MOFs show the ability to association a porous structure with extraordinary drug loading over various kinds of interactions. Therefore, MOFs are perfectly suited as in vivo DDSs. In the past decades, the number of reviews and research papers dedicated to MOFs, Figure 2A, and MOFs and DDSs, Figure 2B, has experienced a steady increase, responding to an emergent request for such smart materials.

A wide variety of factors influence the drug delivery of various MOFs. Besides these factors, we also need to consider the physiochemical properties of MOF together with drug molecule properties, which permit fitting drugs inside the carrier MOF for effective delivery to the desired position. Compared to other molecules that act as drug delivery systems, the mechanism of MOFs drug delivery permits a slow and manageable liberation of drug molecules [50].

Some model examples of drugs delivered by MOFs are presented in the following review, such as: indocianine green (ICG), doxorubicin hydrochloride (DOX), 5-fluorouracil (5-Fu), caffeine, cidofovir, folic acid (FA), calcein (cal) curcumin (Cur), and paracetamol, together with the synthetic classical and new strategies, biocompatibility, and versatility of loading capabilities, Figure 3. 

Additionally, the stimuli and multistimuli responsive variety and type of cancer cell used were considered. Stimuli are used to control how drug delivery systems and drug release act and function. Two types of stimuli are explored: *endogenous stimuli* features of the pathological sites: pH or temperature of local sites, redox status, and hypoxia, and *exogenous stimuli*: ultrasound irradiation, temperature, light, and magnetic fields.

## 2. Modification of MOFs—Cargo Loading Strategies

Wang and coworkers represent three cargo loading strategies: encapsulation, direct assembly, and post-synthesis strategy, to load MOFs with larger amounts of drugs, exemplified in Figure 4. The classification of strategies was made taking into account the location of the drug and the effect of cargo and host–guest interactions within the MOF frameworks. 

*Encapsulation strategy*: Cargo is situated inside of the MOFs pores or channels by *noncovalent* bonding interaction. This type of strategy does not alter the MOF framework structures.*Direct assembly strategy*: The interaction between MOFs and cargo is coordination bonds. The cargo participate in the synthetic reaction as ligands to partly contribute to the construction of MOFs.*Post-synthesis strategy*: The cargo molecules are located in MOFs surfaces. These molecules serve as linkers for pre-synthesized MOFs. The chemical interactions implied by this strategy are coordination bonds and covalent bonds between metal nodes/organic linkers and used cargo. This type of strategy does not alter the MOF framework structures. The second possibility as a post-synthesis strategy is adsorption on MOF surfaces. In general, the main forces in adsorption are weak interactions, such as π–π and Van der Waals interactions, and hydrogen bonding.

### 2.1. Encapsulation Strategy

An extensive series of hydrophilic, hydrophobic, and amphiphilic drug molecules were encapsulated in the MOFs. The library of investigated MOF included in this review—MIL-53(Fe), MIL-53-NH2 (Fe),MOF-74 (Fe),MOF-74 (Zn), MIL-101 (Cr), MIL-100 (Fe), MIL-89 (Fe), MIL-101-NH2 (Fe), MIL-100 (Fe), UiO-66(Zr), MIL-127 (Fe), MIL-100 (Fe), Mi-UiO-68, MIL-88@ZIF-8, ZIF-8, UiO-66-NH2, MIL-88A(Fe)—use various metal–organic framework templates, such as HKUST, UiO, ZIF, MIL, and MOF-78. The MOFs investigated have various stabilities in biological environments.

For biomedical and pharmaceutical applications, the functionalization of MOFs represents a new area of research for binding therapeutic molecules on MOFs surfaces. However, the desired modification must not only reduce the interaction and increase the stability of MOFs with the biologic medium, but it must also simplify the drugs’ passage through the physiological barriers in order to reach the targeted delivery center. Numerous ways were investigated to modify the surface of MOFs materials by optimizing the modification conditions and applying diverse biomolecules, polymers, and ligands, among other molecules [52,53,54]. Recall the excellent physical and chemical properties of MOFs. 

Various parameters, such as pH value, an optimal buffer, nanoparticle size, and surface adaptation, are very important, in addition to analytical approaches and methodologies appropriate to control the MOF stability. The chemical stability of an MOF is unstated as the capacity to keep the long-ranged ordered structure when the material is exposed to a specific environment. Normally, there are two central ways of measuring material degradation: decomposition to the original building blocks (in this way, the components can supplementarily react with various compounds existing in the environment) and amorphization. The chemical stability of MOFs is explained on the basis of pH, time, and temperature.

A large number of researchers combined their efforts to explain MOF stability in acids and bases. In an acidic environment, the MOF degradation is principally caused by competing protons and metal ions trying to coordinate with the organic ligand. In a basic environment, MOF decomposition involves the replacement of organic ligands by hydroxide ions. Consequently, MOFs based on high-valent metal ions and carboxylate ligands are estimated to be rather stable in acids and less resistant to bases. Inversely, the MOFs based on soft divalent metal ions and nitrogen-based ligands are expected to be more stable in basic solutions and further unstable in acids. 

The chemical stability of MOFs and other materials used as drug delivery systems as a function of time is a very important factor to be investigated. Optimal stability is influenced by the application and route of drug administration. Depending on applications, the time range can vary from several days to a few hours, which might be sufficient. As a target, 24 h should be implemented as the initial treatment time, with the possibility of extended times depending on the initial results and needs. For applications in nanomedicine, the preferred temperature is 37 °C in studies involving the chemical stability of MOFs [55]. 

Demel and colleagues explored the stability of UiO-66 MOF in four different buffers: N-ethylmorpholine (NEM), phosphate buffer (PB), 4-(2-hydroxyethyl) piperazine-1-ethane sulfonic acid (HEPES), and 2-amino-2-(hydroxymethyl)-1,3-propanediol (TRIS). The work demonstrated that the fate of zirconium is a major concern in UiO-66 stability. TRIS and PB buffers, depending on concentration, can act as ligands to coordinate with zirconium ions. The kinetics of terephthalate release are accelerated by increasing the temperature between 25 and 37 degrees Celsius [56].

Particle size is an important property of a nanocarrier. Depending on the intended application and the route of administration, different MOF particle sizes are preferred [57]. Particles ranging from 1 to 100 nm exhibit distinctive physicochemical properties. This size allows them to be useful as drug carriers, therapeutics, and profound tissue contrast agents [58]. For instance, nanoparticles smaller than 200 nm are believed to be capable of passive targeting in cancer therapy due to their enhanced permeability and retention effect [59].

To conclude, in gastrointestinal treatments, the mesh pore arrangement of the intestinal mucosa wall (50–1800 nm) works as a sieve that defines which particles are absorbed [60]. The size of MOFs and various NPs used as DDS influenced movement half-life and tumor growth.

Researchers have validated Lewis acid catalytic activity stemming from open metal sites in MOFs. By carefully choosing MOF materials that possesses open metal sites and functionalizing an organic ligand with an organic basic group, in order to obtain materials that present both basic and acidic functional sites [61].

The surface modification of MOFs aids in improving their water stability and drug-loading enrichment, thus altering the degradation pattern and controlling the release of drug molecules.

Indocyanine green (ICG), a water-soluble photosensitizer dye permitted by the US Food and Drug Administration (US FDA) for clinical usage, is being broadly investigated in phototherapy, together with iron oxide nanoparticles—the only nanoparticles approved for clinical practice [62]. As a complex molecule composed of two hydrophobic polycyclic parts holding two additional sulfonate anions at each terminal site (Figure 3a), ICG is capable of interacting with several metal ions, such as the common coordination metal ions Zn^2+^, Mn^2+^, Fe^3+^, and Pt^2+^, and additional trace metals that take part in maintaining homeostasis [63,64]. In the near-infrared (NIR) region, ICG displays light absorption (λabs, max  =  780 nm) and fluorescence emission (λem, max  =  800 nm) applied to biologically transparent systems [65]. Yang and coworkers investigated one-step encapsulation of ICG in ZIF-8 nanoparticles (NPs). The encapsulation was evidenced using near infrared because ICG@ZIF-8 possesses an absorption band at 783 nm and possesses high photothermal conversion efficiency. In vivo and in vitro studies were performed which reveal that ICG@ZIF-8 after loading DOX forms ICG@ZIF-8-DOX NPs, which display the chemical and photothermal synergistic treatment for tumors [66]. In some cases, ICG effectiveness is limited by photobleaching and its tissue selectivity is low. 

Doxorubicin hydrochloride (DOX) is the first line chemotherapeutic drug for various types of cancer such as lymphoblastic leukemia, ovarian cancer, and breast cancer [67,68], Figure 3b.

Yao and coworkers synthesized a water-stable MIL-101(Fe)-C_4_H_4_ via a microwave-assisted technique starting from iron (III) chloride hexahydrate and naphthalene-1,4-dicarboxylic acid. The nano-MOFs operated as sponges when saturated in DOX alkaline aqueous solution with an experimental loading capacity of up to 24.5 wt%, with and an upholding loading efficiency of about 98% [69].

5-Fluorouracil (5-Fu), a pyrimidine analogue (Figure 3c), has been an important anticancer agent for managing an extensive diversity of tumors because it can be combined with DNA and RNA, leading to cancer cell death. 5-Fu, a small-molecular-weight drug, was investigated and used in biomedical applications as a model anticancer drug [70,71].

Caffeine, Figure 3d, is a liporeductor and an amphiphilic drug, with an effective one molecule volume of 165.5 Å^3^. Caffeine molecules were often investigated as a model active component; many preparations of caffeine were contained within numerous diverse matrices. The combination with MOFs makes these cosmetic-containing composite MOF-based patches encouraging nominees for new devices applied on the skin and used in cosmetic applications [72]. Liedana and coworkers investigated one-pot encapsulation of caffeine drugs in ZIF-8 a zeolitic imidazolate MOF [73]. This one-pot strategy reveals a larger quantity of caffeine encapsulated when compared with the encapsulation in two steps, which was presented by Gaudin and co-workers [74] using MIL-88B_2OH as a drug delivery system. Furthermore, by comparing the two MOFs as caffeine drug delivery systems, it was observed that ZIF-8 provides a slower delivery, most likely due to the strong interaction formed between caffeine and ZIF-8.

With an excellent affinity for the folate receptors, folic acid (FA, Figure 3f) is used in combination with various MOFs as drug carriers to reduce the toxicity of anticancer drugs to normal cells and increase the anticancer effects on cancer cells [75,76]. In most cases, the folic acid receptors are overstretched on the external surface of cancer cells. Therefore, FA is broadly used as a model targeting drug for proficiency in delivering drugs into tumor cells [77]. Owing to its high stability and good compatibility in physiological environments, FA acts as a perfect targeting molecule for delivery systems.

Calceine (cal), a hydrophylic fluorescent molecule, nontoxic to the cell, which can be traced by confocal microscopy, was encapsulated by Orellana-Tavra and coworkers in UiO-66 MOF as DDS, forming cal@UiO-66 MOF. UiO-66 has a cubic structure with two types of cavities (ca. 11 and 8 Å and uses Zr—a low toxicity metal). The material was ball-milled, forming an amorphous material named cal@a_m_UiO-66 MOF. The ball-milling amorphization process allows the entrapment of the calceine inside the porosity. This process significantly increases drug release times from 2 days (cal@UiO-66 MOF) up to 30 days (cal@amUiO-66) in a drug delivery system (DDS) [78]. Calcein-loaded MOFs triggered higher cargo internalization related to the free molecule of calcein; enhanced cargo internalization, minimizing undesired side effects. Cal@MIL-101(Cr) outperformed Cal@UiO-66 at concentrations higher than 10 mg m L^−1^, while the calcein transport efficacy was appropriate at 1 mg mL^−1^ with an enriched uptake related to free calcein, which was higher after 24 h [79].

Over the past decades, curcumin, the active constituent of the *Curcuma longa* plant, has received great consideration as an anti-inflammatory agent, antioxidant, and anticancer representative [80]. Metal–organic frameworks (MOFs) are employed as possible drug-delivery systems for controlling the pharmacokinetic rate of drug delivery. Combining two MOF-74 (a solid with a pore size of 12 nm, designed by combining 2,5-dihydroxyterephthalic acid with divalent metal ions Mg^2+^ and Zn^2+^ (Mg-MOF-74 and Zn-MOF-74) with different solubility of the metal nodes, Tomeh and coworkers investigated the pharmacokinetic eliberation rate of curcumin as a drug. In the study, the ratio of Mg-MOF-74 and Zn-MOF-74 was modified to the following limits: 80:20, 60:40, 40:60, and 20:80 wt% Mg:Zn to control the pharmacokinetic concern at a rate of 30 wt% curcumin. The drug delivery trials were performed in saline phosphate buffered by varying the time from 0 to 24 h, and keeping the temperature at 37.4 °C. The revealed amount of curcumin was found to be higher when the concentration of Mg-MOF-74 was higher. The drug concentration heightened both the raw delivery and the pharmacokinetic rate of drug release. 

Gao and coworkers [81] fabricated a nanoplatform with extended circulating properties. UiO-66, a zirconium-based MOF, was used as a transporter for O_2_ storage. UiO-66 was linked with indocyanine green (ICG) as a first step by coordination reaction, followed by surface modification by covering with red blood cell (RBC) membranes, Figure 5.

C. Chu and co-workers [82] investigated zinc(II) coordination-based nanoformulations (zinc(II)-dipicolylamine) adept at loading indocyanine green (ICG) and therapeutic genes. For various therapy strategies, such as fluorescence behavior combined with photoacoustic-imaging-guided photo/gene therapy and zinc(II)-dipicolylamine-based photosensitizer metal–organic nanostructures, were synthesized to accomplish various therapy strategies. Considerable work has been performed in this direction and valuable research outcomes have been evidenced. Nevertheless, challenges still remain for clinical interpretation, and because of this some important factors are still unsatisfied, such as biocompatibility, limited efficacy, and complex procedure of synthesis.

Fang and colleagues [83] published a useful study of a versatile nanoplatform composed of core-shell ZnS@ZIF-8 nanoparticles combined with ICG and tirapazamine (TPZ), denoted as ZnS@ZIF-8/ICG/TPZ (ZSZIT), designed and prepared to enable H_2_S-sensitized chemo- or photodynamic synergistic therapy. The degradation of core–shell structure and H_2_S release properties are important features of nanoparticles for the antitumor purposes. The study exhibit a considerable anticancer effect, in vitro and in vivo, produced by the joint effects of intracellular reactive oxygen species, H_2_S, and activated TPZ empowered by ZnS@ZIF-8/ICG/TPZ nanoparticles.

You and coworkers studied the effect of a Pt-based oxygen nanogenerator UiO-66-like MOF [84], which can continuously use endogenous H_2_O_2_ inside the tumor microenvironment of the solid tumor by the contained Pt nanozyme, enriching the oxygen concentration in situ, which considerably improved tumor hypoxia and supplementary heightened the effect of photodynamic therapy. The stable, porous Zr-BDC—a sustainable substitute to the conventional UiO-66(Zr) MOF—enables the loading of large volumes of Pt nanozymes.

Fan and coworkers made a multimodal imaging-led synergistic cancer photoimmunotherapy by using MIL101-NH_2_ dual-dressed with photoacoustic and fluorescent indicator donors ICG and immune adjuvants (CpG ODN- cytosine-phosphate-guanine sequence-oligo nucleotides) as the core transporter. The strategy behind this material is to use it to tune cold tumors to hot. When light activates the photothermal therapy, the immune system is triggered to show a long-acting antitumor result. The dual-dressed MIL101-NH_2_ brings a favorable method used for the prognosis of cancer and its treatment [85].

Wu and coworkers constructed a photosensitizer, ICG, and a chemotherapy-induced drug, DOX, which were encapsulated step by step inside the nanopores of the MIL-88 core and ZIF-8 shell to exhibit a synergistic photothermal, photodynamic, and chemotherapy nanoplatform. Both coreshell MOFs, MIL-88, and ZIF-8 have two functional regions for codrug delivery. DOX relief could be triggered by a low pH value and the speed of this effect is increased by near-infrared light irradiation [86]. 

Su and coworkers [87] encapsulated curcumin and ICG into ZIF-8 pores in a one-pot synthesis method and synthesized the compound entitled ICG&Cur@ZIF-8, which exhibits pH-controlled drug delivery and good photothermal performance. The codelivery of two drugs at the same time opens the way to new opportunities for biomedical application. The percentage of drug loading capacity was calculated using the experimental UV–VIS spectra, at 9.6 wt% for Cur and 12.3 wt% for ICG, respectively. 

### 2.2. Direct Assembly Strategy

Magnetic nanoparticles of MOF act as a drug delivery system by directing the cure treatment location without disturbing the other cells of the body. In this respect, Gautam and coworkers synthesized a porous and flexible Cu-based (MOF), Cu-benzene tricarboxylic acid, well known as HKUST-1. The crystalline growth of Cu-BTC has been optimized using a hydrothermal technique. A selection of nontoxic solvents was used in the synthesis [88]. The absorption of paracetamol medication on HKUST-1 at 10 h and 48 h is notable.

Yang and coworkers [89] fabricated a nano-MOF by using a direct assembly strategy. In order to test the compound as photothermal therapy, a MOF composed of Mn^2+^ and IR825 near-infrared (NIR) dye was synthesized. The prepared compound was named Mn-IR825. The first step was to coat it with a polydopamine (PDA) shell. The last step was to functionalize it with polyethylene glycol (PEG), thus becoming Mn-IR825@PDA−PEG, a compound with a great photostability. The DDS offers strong contrast in T1-weighted MRI, IR825 with strong NIR optical absorbance. The tests were performed on mice and the Mn-IR825@PDA−PEG showed rapid renal excretion, thus decreasing the long-term toxicity effect. During laser irradiation at 808 nm, tumors in mice with Mn-IR825@PDA−PEG injection were totally removed without reappearance within 60 days. Therefore, Mn-IR825@PDA−PEG is a good candidate for applications in cancer theranostics.

Lu and coworkers [90] produced the first chlorin-based metal−organic framework nanomaterial named DBC-UiO, which was used to kill two colon cancer in cell mouse models by applying DBC-UiO-induced photodynamic therapy (PDT).

Rabiee and coworkers [91] used two-dimensional MOF-5 inserted MXene nanostructures in the codelivery of DOX and to increase its bioavailability; their interaction with the pCRISPR was studied. The MXene/MOF-5 was coated with *alginate* and *chitosan.* Cytotoxicity tests were conducted on different cell lines such as HeLa, PC12, HEK-293, and HepG2. The results emphasize the outstanding cell viability at very low and high concentrations. *The chitosan-coated MOFs* presented higher relative cell viability, more than 60% in all cell lines investigated. *The alginate-coated* MOFs placed second in cell viability, with approximately 50%. Additionally, a significant drug delivery of 35.7% was accomplished, due to the interactions between the MXene/mMOF-5 and the doxorubicin drug.

Sun and coworkers investigated one porphyrinic MOF nanoplatform (H-PMOF) and achieved enhanced photodynamic therapy effectiveness related to nonhollow MOF nanoparticles. Additionally, H-PMOF was investigated as a drug delivery system to coload DOX and indocyanine green (ICG) with an excellent drug-loading ability of 635%. Additionally, cancer cell tissue cover-up of the (DOX and ICG)@H-PMOF composite nanoparticles gave a biomimetic nanoplatform, namely (DOX and ICG)@H-PMOF@mem, with a remaining tumor-targeting and immune-escaping capacity. The following multistimuli: near-infrared laser-triggering and pH-controlled were used for DOX release from (DOX and ICG)@H-PM OF@mem. MOF-based multifunctional nanomedicine development and design are being considered for the combination of cancer healing and specific theranostics [92].

### 2.3. Post-Synthesis Strategy

Alves and coworkers [93] applied a post-synthesis strategy to reach a tumor-targeting drug-loaded MOF using “click” chemistry. As a starting MOF, N3-bio-MOF-100 was used and folic acid molecules were anchored on its surface. The anticancer model drug used in the investigations was curcumin, measuring the drug-loading encapsulation effectiveness at 24.02 percent after soaking the MOF for 1 day in a curcumin solution and 25.64% after 3 days of soaking. The synthesized MOF DDS was successfully characterized by ^1^H-NMR, FTIR, and LC–MS. The N3-bio-MOF-100 post-synthesized was investigated as a stimuli-responsive drug release. The studies were performed on the breast cancer 4T1 cell line. Curcumin has been shown to increase in release in acidic microenvironments.

Yang and coworkers used a post-synthetic modification of nanoscale zinc MOFs (nanoMOFs) functionalized with folate to build tumor cell-targeted material, named FA-IRMOF-3 by conjugating the -NH_2_ group of IRMOF-3 with the -COOH groups of folic acid. The FA-IRMOF-3 was stocked with 5-Fu by impregnating it with 5.0 mg of drug (5-Fu) for 1, 3, 5, or 7 days. During drug loading tests, nanoMOFs functionalized with folate rapidly adsorb up to 24 wt% of 5-fluorouracil while retaining the FA-IRMOF-3’s crystalline structure. In folate receptor-positive KB and HeLa cells, 5-Fu and 5-Fu-FA-IRMOF-3 cytotoxicity were investigated. The results showed that 5-Fu exhibited higher cytotoxicity than 5-Fu-IRMOF-3, and the cytotoxicity of 5-Fu-FA-IRMOF-3 was stronger compared with free 5-Fu [94]. 

The literature assessment of MOF-based nanomaterials for DDS is presented in Table 1.

## 3. Stimuli Responsive for MOFs as DDSs

Cai and coworkers [110] explained the basic mechanisms of stimuli-responsive DDSs described in Figure 6. After injection, the stimuli-responsive drug delivery system (MOFs or other types of nanomaterials) gathers at the tumor site and is activated in two ways: one enhanced by permeability and preservation effect, and the other one enhanced by receptor–ligand affinity. When stimulated by specific triggers, the DDSs undergo a transformation, which induces drug delivery at the planned site in a controlled manner. Paralleled with healthy cells, the tumor cells microenvironment has different characteristics: higher temperature or lower pH, which helps to develop internal stimuli-responsive DDS. The external (light, magnetic field, temperature, ultrasound irradiation) stimuli-responsive systems envisage the MOFs gathered at the tumor site and are activated by triggers placed outside the body. The external stimuli-responsive systems offer improved continuous drug release capacity and have established a higher potential for biomedical application. 

One way to unlock MOFs and relieve their loads is through the following method: the loaded MOFs with chemical functionalities attached or absorbed on their surface are secured, acting as entrances for the activated release of the loads [111,112,113,114]. Among the MOF nanocarriers generated by outside stimuli, due to the acidic tumor microenvironment, the pH-responsive MOFs are the most broadly investigated [115,116,117]. MOFs based on iron [118], zinc [119,120,121], zirconium [122,123], gadolinium [124], hafnium [125], and europium [126] were investigated for pH responsive stimuli in tumor treatment.

Post-synthetic modification in MOFs, particularly cation exchange, has been a highly explored research area due to its potential applications in various domains, providing novel functional materials. Hamisu and coworkers [127] thoroughly described how the hard–soft acid–base (HSAB) principle permits an overabundance of experimental clarifications that manage cation exchange at the secondary building units (SBUs) of MOFs.

The strength or weakness of MOFs can be affected by numerous factors, such as: interaction of metal ions and organic linkers that describes the strength of the M–L bonds, coordination geometry, pore surface properties, and working environments [128,129]. The HSAB principle is appropriate to define a moderately stable MOF system, in which a soft acid binds more strongly to a soft base, and a hard acid has a preference for a hard base. MOFs prepared from borderline divalent cations and azolate linkers, such as the ZIF-based MOFs ZIF-8 [130], ZIF-71 [131], and ZIF-90 [132], are found to be the most stable for a wide range of pH and a large variety of solvents.

In this respect, stable MOFs, based on hard–hard interactions, imply high valent cations and carboxylate linkers, or those based on soft–soft interaction, the case of low valent cations and azolate linkers, which are more challenging to break and interchange due to strong M–L bonds. In this case, external energy is necessary to overcome the bond dissociation energy and to facilitate the cation exchange.

To achieve improved therapeutic efficacy, supplementary efforts should be dedicated to the development of MOF-based, multiple stimuli-responsive drug delivery systems.

## 4. Conclusions and Future Perspectives

In conclusion, we studied the recent significant progress in nanometal–organic framework. The library of MOFs investigated in this review included MIL-53(Fe), MIL-53-NH2 (Fe),MOF-74 (Fe),MOF-74 (Zn), MIL-101 (Cr), MIL-100 (Fe), MIL-89 (Fe), MIL-101-NH2 (Fe), MIL-100 (Fe), UiO-66(Zr), MIL-127 (Fe), MIL-100 (Fe), Mi-UiO-68, MIL-88@ZIF-8, ZIF-8, UiO-66-NH_2_, and MIL-88A(Fe), which use various metal–organic framework templates, such as HKUST, UiO, ZIF, MIL, and MOF-78 as drug delivery systems, and categorized the cargo delivery strategies into three types: encapsulation, direct assembly as a classical approach, and post-synthesis strategy as a new approach. The MOFs investigated have different stabilities in biological surroundings. The surface and available anchorages for a large number of cargos—indocianine green (ICG), doxorubicin hydrochloride (DOX), 5-fluorouracil (5-Fu), caffeine, cidofovir, acid folic (FA), calcein, curcumin and paracetamol—are well established. In comparison to commonly used cargo carriers such as carbon nanotubes, graphenes, magnetic nanoparticles, biomolecules, polymers, inorganic silica, and gold nanoparticles, the study of MOF and its nanoscale development are being investigated using both experimental and computational approaches based on cargo loading and release within the pores of MOF at the atomic level. Among the various classes of NPs compounds used as DDSs, MOFs are considered an encouraging class of nanocarriers for DDSs, exhibiting a well-defined arrangement, extraordinary surface area and porosity, versatile pore size, and easy chemical functionalization. Comparing the two types of stimuli, the external responsive stimuli have greater flexibility. Each stimulus-responsive modality has its own limitations.

However, by surveying the current investigation, there are still numerous tasks that need to be accomplished. Taking into account the biomedical application, it is necessary to carefully investigate the toxicity and solubility of the selected building blocks and metal nodes for synthesized MOFs, as well as the biocompatibility issues. In addition to toxicity, efforts should be made to improve the size and morphology of MOF nanoparticles, to certify the prolonged and retarded blood circulation, and to precisely control the drug release and target the tumor after drug administration. Notwithstanding these challenges, the bioapplication of MOF nanoparticles has significantly improved since the first report on MOF for drug loading in 2006, and MOFs as DDS are expected to achieve extensive advances in nanomedicine.

## Figures and Tables

**Figure 1 ijms-23-04458-f001:**
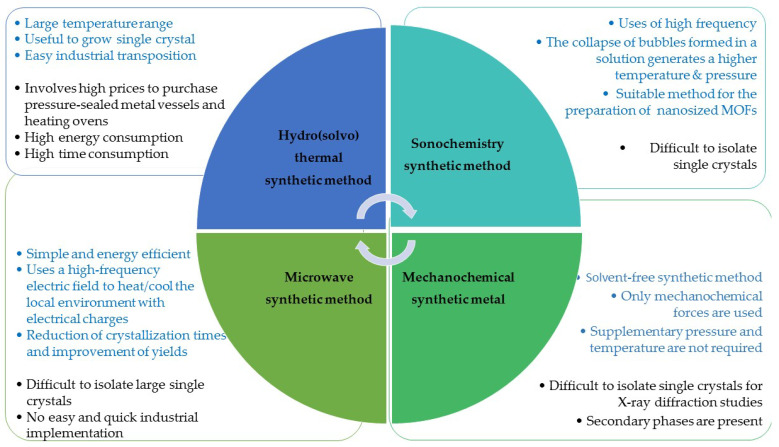
MOFs synthesis—alternative reaction conditions. Features shown in blue are advantages and black are disadvantages for all described methods.

**Figure 2 ijms-23-04458-f002:**
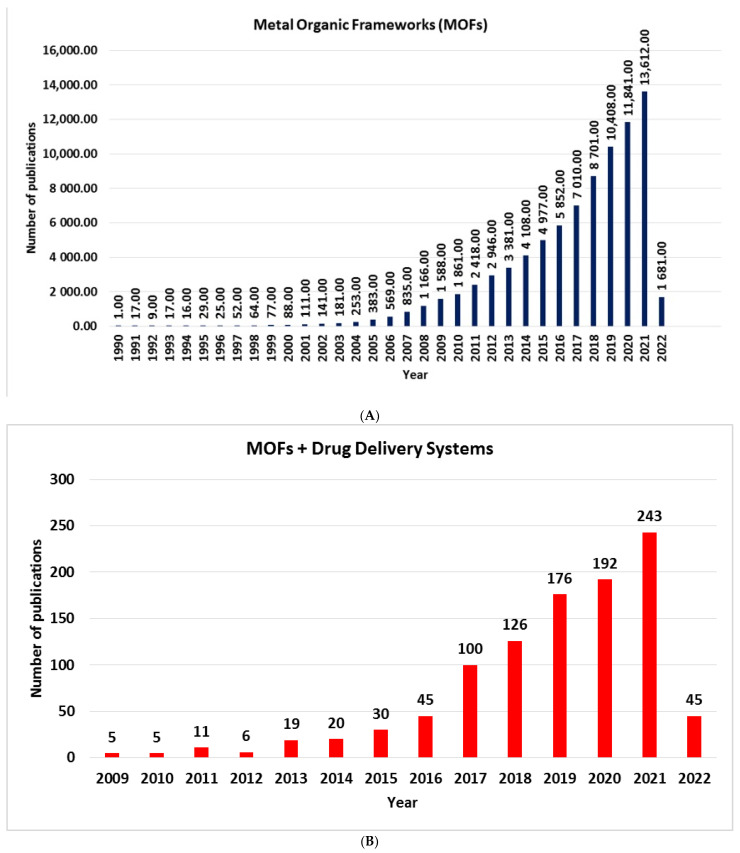
Number of publications from Web of Science: (**A**) “Metal Organic Frameworks” and (**B**) “Metal Organic Frameworks and Drug Delivery Systems”, from 1996 and 2009, through 21 March 2022.

**Figure 3 ijms-23-04458-f003:**
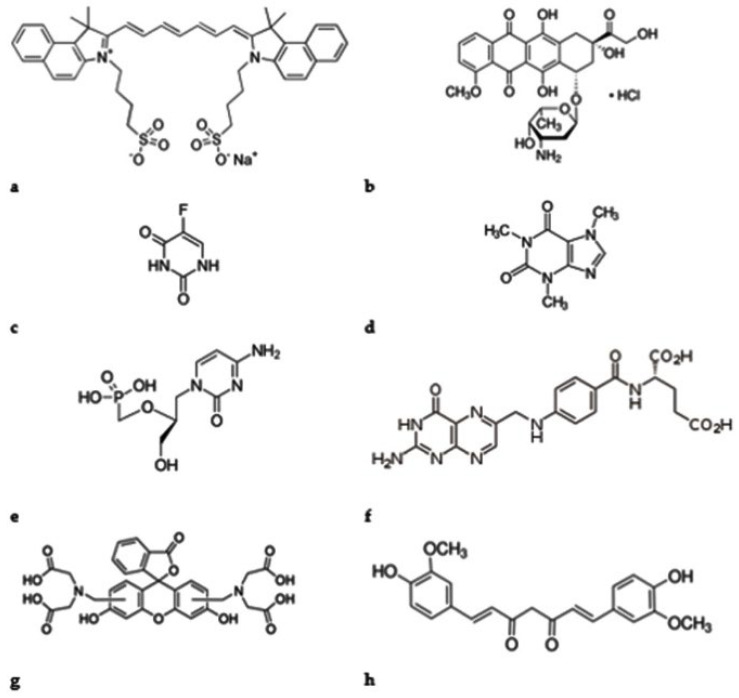
Some examples of drugs delivered by MOFs as DDS: (**a**)—indocianine green, (**b**)—doxorubicin hydrochloride, (**c**)—5-fluorouracil, (**d**)—caffeine, (**e**)—cidofovir, (**f**)—acid folic, (**g**)—calcein, (**h**)—curcumin.

**Figure 4 ijms-23-04458-f004:**
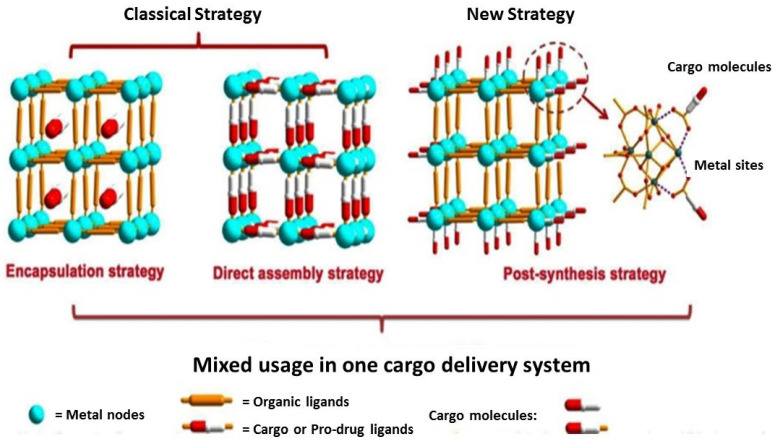
Encapsulation, direct assembly, and post-synthesis of cargo-loading strategies for MOFs. Reproduced with permission from [51]; Copyright (Wang, 2018), *RSC J. Mater. Chem. B*.

**Figure 5 ijms-23-04458-f005:**
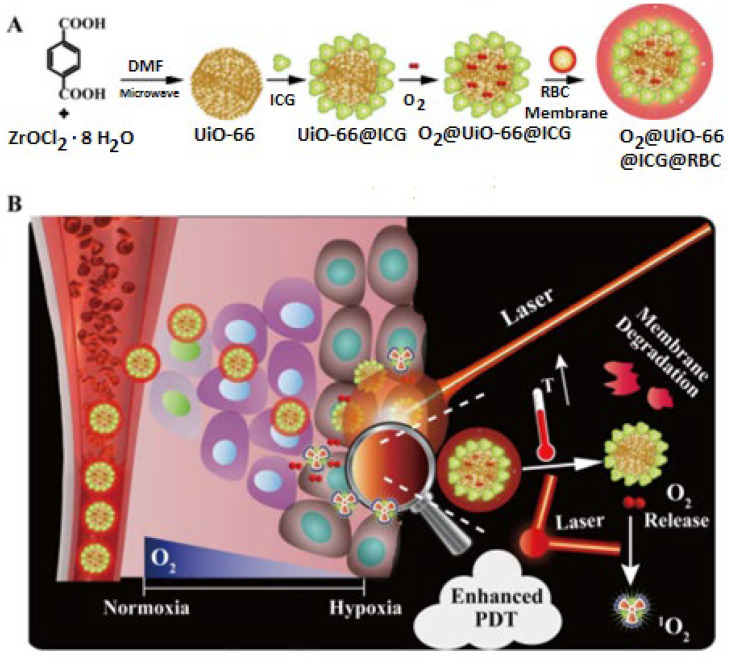
Synthesis of O_2_@UiO-66@ICG@RBC (**A**). Schematic mechanism of NIR-triggered O_2_ enhanced and discharging PDT (**B**). Reproduced with permission from [81]; Copyright (Gao, 2018), Elsevier Biomaterials.

**Figure 6 ijms-23-04458-f006:**
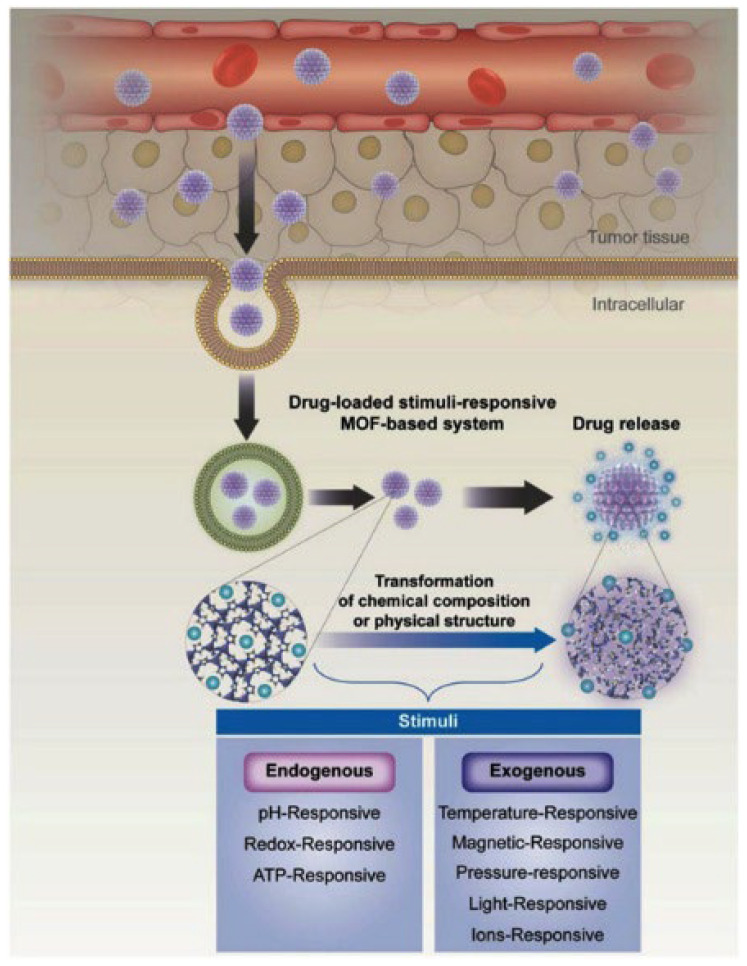
Schematic illustration of metal–organic frameworks (MOFs)-based stimuli-responsive system for drug delivery. Reproduced from reference [110]; Copyright (Cai, 2018) Wiley Online Library.

**Table 1 ijms-23-04458-t001:** A literature assessment of MOF-based nanomaterials for drug delivery systems.

MOFs	Components		Pore Size or Volume/Particle Size	Therapeutic Agent	Drug Loading Percentage	Outcome	Ref.
	Organic	Inorganic					
MIL-53(Fe)	Terephtalic acid	FeCl_3_·6H_2_O	8.6 Å/ 350 nm	Ibuprofen Caffeine	22 23.1	antitumoral and retroviral drugs against cancer and AIDS	[95]
MIL-53-NH_2_ (Fe)	2-Amino-terephtalic acid	FeCl_3_·6H_2_O	120 nm	5-Fu	28	magnetic resonance, optical imaging, targeted drug delivery	[96]
MOF-74-Fe	1.4-Dihydroxy terephtalic acid	FeCl_2_·4H_2_O	-/200–800 nm × some μm	Ibuprofen	15.9	low cytotoxicity, efficient drug loading capacity, controllable drug release	[97]
MOF-74-Zn	2.5-Dihydroxyterephthalic acid	Zn(NO_3_)_2_ 6H_2_O	12.7 A/20 nm	Ibuprofen	50	fast kinetics (k = 0.27 h^−1/2^), high drug conc. in first 10 h.	[98]
MIL-101 (Cr)	1,4-benzenedicarboxylates,	trimeric chromium(III) octahedral clusters	34 Å	Ibuprofen	-	can adsorb 138 wt% ibuprofen, release the total amount slowly in 6 days	[99]
MIL-100 (Fe)	Trimesic acid	Fe (NO_3_)_3_·9H_2_O or FeCl_3_·6H_2_O	29 A/200 nm	Ibuprofen urcumin	33	antitumoural and retroviral drugs, photoacoustic imaging-guided chemo-photothermal combinational tumor therapy	[95,100,101]
MIL-89 (Fe)	Muconic acid	FeCl_3_·6H_2_O	11A/50–100 nm	Cidofovir	14	antitumoral and retroviral drugs	[95]
MIL-101-NH_2_ (Fe)	Aminoterephtalic acid	FeCl_3_·6H_2_O	34 A/120 nm	Cidofovir	41.9		[95]
MIL-100 (Fe)	Trimesic acid	Fe (NO_3_)_3_·9H_2_O or FeCl_3_·6H_2_O	1.2 cm^3^/g /102.8 nm	DOX Caffeine	28 24.2	released DOX in a pH-dependent manner, breast cancer treatment	[102]
UiO-66(Zr)	1,4-Benzenedicarboxylic acid	ZrCl_4_	5–7 A	Caffeine DOX	22.4 15.1	caffeine molecules are preferentially located in the smaller cages, DOX-containing PEGylated nanoMOFs exhibit notable redispersibility	[103,104]
MIL-127 Fe	3,3′,5,5′-Azobenzene tetracarboxylate	trimers of iron(III) octahedra	4 A	Caffeine	15.9	carriers for topical administration of caffeine	[105]
MIL-100 (Fe)	Trimesic acid	{Fe_3_O} trimer	200 nm	DOX	9.1	full release of drug in 5 days	[95]
Mi-UiO-68	maleimide-attached H_2_L ligand	ZrCl_4_	25.6 A	DOX FA (acid folic)	4.84	multifunctional cancer treatment system	[106]
MIL-88@ZIF-8	BDC-NH_2_, 2-Me-IM	FeCl_3_·6H_2_O Zn(NO_3_)_2_·6H_2_O	-/1.3 nm	ICG DOX	3.58% 21.69%	MIL-88-ICG@ZIF-8-DOX core-shell dual MOF for synergistic cancer, photothermal and photodynamic therapy	[86]
ZIF-8	Me-IM	Zn(NO_3_)_2_·6H_2_O	200 nm	5-Fu	21.2	biological purposes: bio–nano interaction, pulmonary accumulation, antitumor therapy	[107]
UiO-66-NH_2_	NH_2_-BDC	{Zr_6_O_8_} cluster	100 nm	5-Fu	3.1	multistimuli responsive in bone diseases: increasing Ca^2+^ concentration, decreasing pH, thermal therapy	[108]
MIL-88A (Fe)	Fumaric acid	FeCl_3_·6H_2_O	6 A/150 nm	Cidofovir	2.6		[109]

## Data Availability

Not applicable.

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
