# Peer review of "Applications of Metal-Organic Frameworks as Drug Delivery Systems"

_ijms, 2022, doi:10.3390/ijms23084458_

Round 1
Reviewer 1 Report
This work can be assumed as a complete and comprehensive one. The authors put a lot of organized effort in order to present the recent trends on utilization of MOFs as drug delivery systems. I would suggest publication after:
- Polishing the Figure and increase their quality/resolution.
- Fixing some minor linguistic issues
- Adding the potential of building composites of MOFs (example: 10.3390/molecules25030513) and discuss slightly more the HSAB theory of stability and how can be linked with the bio/body-world.
Author Response
This work can be assumed as a complete and comprehensive one. The authors put a lot of organized effort in order to present the recent trends on utilization of MOFs as drug delivery systems. I would suggest publication after:
Answer: Thank you very much for your recommendations and the new reference. We apologize for these mistakes.
- Polishing the Figure and increase their quality/resolution.
Answer: The figures were polish and their quality/resolution was increased.
- Fixing some minor linguistic issues
Answer: Thank you very much. The minor linguistic mistakes were found and resolved.
- Adding the potential of building composites of MOFs (example: 10.3390/molecules25030513) and discuss slightly more the HSAB theory of stability and how can be linked with the bio/body-world.
Answer: We were analyzing new references, and we inserted the new information provided by the abovementioned paper and new ones related to the subject as follows:
Post synthetic modification in MOFs, particularly cation exchange, has been a highly explored research area due to its potential applications in various domains, providing novel functional materials. Hamisu and coworkers [128], described extensively how the Hard-Soft Acid-Base (HSAB) principle, permits an overabundance of experimental clarifications dealing with cation exchange at the Secondary building units (SBUs) of MOFs.
The strength or weakness of MOFs can be affected by numerous factors, such as: the interaction of the metal ions and the organic linkers, which describes the strength of the M–L bonds, coordination geometry, pore surface properties, and working environments [129, 130]. The HSAB principle is appropriate to define a moderately stable MOF system, in which a soft acid binds more strongly to a soft base and a hard acid has a preference for a hard base. MOFs prepared from borderline divalent cations and azolate linkers such as ZIF based MOFs: ZIF-8 [131], ZIF-71 [132] or ZIF-90 [133] are found to be the most stable, in a widespread pH and solvents variety.
In this respect, stable MOFs, based on hard-hard interactions, imply high valent cations and carboxylate linkers or those based on soft-soft interaction, the case of low valent cations and azolate linkers, they are more challenging to break and interchange due to strong M–L bonds. In this case, external energy is necessary to overcome the bond dissociation energy and to facilitate the cation exchange.
128 Hamisu, A.M.; Ariffin, A.; Wibowo, A.C. Cation Exchange in Metal-Organic Frameworks (MOFs): The Hard-Soft Acid-Base (HSAB) Principle Appraisal, Inorganica Chim. Acta 2020, doi: https://doi.org/10.1016/j.ica.2020.119801
- Vardali, S.C.; Manousi, N.; Barczak, M.; Giannakoudakis, D.A. Novel Approaches Utilizing Metal-Organic Framework Composites for the Extraction of Organic Compounds and Metal Traces from Fish and Seafood. Molecules 2020, 25, 513. https://doi.org/10.3390/molecules25030513
- Wang, C.; Liu, X.; Demir, N.K.; Chen, J.P.; Li, K. Applications of water stable metal–organic frameworks, Chem. Soc. Rev. 2016, 45, 5107-5134
- K.S. Park, Z. Ni, A.P. Côté, J.Y. Choi, R. Huang, F.J. Uribe-Romo, H.K. Chae, M. O’Keeffe, O.M. Yaghi, Exceptional chemical and thermal stability of zeolitic imidazolate frameworks, Proc. Natl. Acad. Sci. 2006, 103, 10186-10191
- Fei, H.; Cahill, J.F.; Prather, K.A.; Cohen, S.M. Tandem postsynthetic metal ion and ligand exchange in zeolitic imidazolate frameworks, Inorg. Chem., 2016, 52, 4011-4016.
- Huang, A.; Dou, W.; Caro, J.R. Steam-Stable Zeolitic Imidazolate Framework ZIF-90 Membrane with Hydrogen Selectivity through Covalent Functionalization J. Am. Chem. Soc. 2010, 132, 15562-15564.
Thank you very much for your valuable comments. The paper was amended in order to improve its quality.
Reviewer 2 Report
The Review entitled “Applications of Metal-Organic Frameworks as Drug Delivery Systems” deals with an interesting area of both Environmental and Agricultural fields. First of All, I want to thanks all the authors for this excellent review article. The writing quality is good too and similarity is very low. The review is very informative, precise, and comprised of relevant content. This paper deserves to be published in IJMS. I strongly recommend this paper for publication.
Abstract
A comprehensively written "Abstract" minutely explains the consequence of research focusing upon vital outcome of importance. Abstract is nicely presented. However, some errors have been detected in the forms of grammar, spell mistakes.
Introduction
The literary structure of the introduction is also good and has been arranged well with suitable information. Wonderfully consulted pertinent literature up to 2020, authors have planned research with scientifically sound hypotheses that consists of several appreciative objectives. There are still some minor mistakes need to be corrected.
- The language has no flaw and it is exceptional. Creditably, a very scientific description of results has been given followed by excellent discussion based on evidence up to the year 2020. Above all, the tables are neatly presented and in conformity of the result description besides satisfactory figures.
Conclusion
- Conclusions are extremely to the point elucidating the novel work and desirable findings.
Fig. photos and tables are well connected with literature and beautifully presented.
References are well written and linked with every feature of the literature. They are up to the mark and there is no missing of reference either in text or reference part.
Author Response
The Review entitled “Applications of Metal-Organic Frameworks as Drug Delivery Systems” deals with an interesting area of both Environmental and Agricultural fields. First of All, I want to thanks all the authors for this excellent review article. The writing quality is good too and similarity is very low. The review is very informative, precise, and comprised of relevant content. This paper deserves to be published in IJMS. I strongly recommend this paper for publication.
Abstract
A comprehensively written "Abstract" minutely explains the consequence of research focusing upon vital outcome of importance. Abstract is nicely presented. However, some errors have been detected in the forms of grammar, spell mistakes.
Introduction
The literary structure of the introduction is also good and has been arranged well with suitable information. Wonderfully consulted pertinent literature up to 2020, authors have planned research with scientifically sound hypotheses that consists of several appreciative objectives. There are still some minor mistakes need to be corrected.
- The language has no flaw and it is exceptional. Creditably, a very scientific description of results has been given followed by excellent discussion based on evidence up to the year 2020. Above all, the tables are neatly presented and in conformity of the result description besides satisfactory figures.
Conclusion
- Conclusions are extremely to the point elucidating the novel work and desirable findings.
Fig. photos and tables are well connected with literature and beautifully presented.
References are well written and linked with every feature of the literature. They are up to the mark and there is no missing of reference either in text or reference part.
Answer
Thank you very much for your valuable comments and appreciation. The paper was amended in order to improve its quality. The small mistakes detected in the form of grammar and spelling were corrected. The quality of the figures was improved.